# Prediction of Poststroke Depression Based on the Outcomes of Machine Learning Algorithms

**DOI:** 10.3390/jcm11082264

**Published:** 2022-04-18

**Authors:** Yeong Hwan Ryu, Seo Young Kim, Tae Uk Kim, Seong Jae Lee, Soo Jun Park, Ho-Youl Jung, Jung Keun Hyun

**Affiliations:** 1Department of Rehabilitation Medicine, College of Medicine, Dankook University, Cheonan 31116, Korea; ryhryh1231@dkuh.co.kr (Y.H.R.); juliet8383@naver.com (S.Y.K.); magnarbor@dankook.ac.kr (T.U.K.); rmlee@dankook.ac.kr (S.J.L.); 2Welfare & Medical ICT Research Department, Electronics and Telecommunications Research Institute, Daejeon 34129, Korea; psj@etri.re.kr; 3Department of Nanobiomedical Science & BK21 NBM Global Research Center for Regenerative Medicine, Dankook University, Cheonan 31116, Korea; 4Institute of Tissue Regeneration Engineering (ITREN), Dankook University, Cheonan 31116, Korea

**Keywords:** poststroke depression, prediction, machine learning, functional scale, cognitive scale

## Abstract

Poststroke depression (PSD) is a major psychiatric disorder that develops after stroke; however, whether PSD treatment improves cognitive and functional impairments is not clearly understood. We reviewed data from 31 subjects with PSD and 34 age-matched controls without PSD; all subjects underwent neurological, cognitive, and functional assessments, including the National Institutes of Health Stroke Scale (NIHSS), the Korean version of the Mini-Mental Status Examination (K-MMSE), computerized neurocognitive test (CNT), the Korean version of the Modified Barthel Index (K-MBI), and functional independence measure (FIM) at admission to the rehabilitation unit in the subacute stage following stroke and 4 weeks after initial assessments. Machine learning methods, such as support vector machine, k-nearest neighbors, random forest, voting ensemble models, and statistical analysis using logistic regression were performed. PSD was successfully predicted using a support vector machine with a radial basis function kernel function (area under curve (AUC) = 0.711, accuracy = 0.700). PSD prognoses could be predicted using a support vector machine linear algorithm (AUC = 0.830, accuracy = 0.771). The statistical method did not have a better AUC than that of machine learning algorithms. We concluded that the occurrence and prognosis of PSD in stroke patients can be predicted effectively based on patients’ cognitive and functional statuses using machine learning algorithms.

## 1. Introduction

Poststroke depression (PSD) is one of the most common psychiatric disorders in stroke patients [1,2]. Its incidence ranges from 10 to 52% according to subject selection or diagnostic criteria [2,3,4,5,6]. The pathophysiology of PSD is not obvious, although it might be associated with the secondary effects of psychological distress and cognitive impairment, not stroke itself [7]. The relationship between PSD and cognitive impairment has already been widely investigated. Cognitive impairment, in fact, is known to be one of the major predictors of PSD [6,8,9]; moreover, PSD is associated with a greater degree of cognitive impairment and has a negative impact on the activities of daily living (ADL) that are integral to recovery [4,5,10,11]. Based on the strong relationship between PSD and cognitive impairment, reductions in PSD symptoms might also enhance patients’ cognitive and functional recovery [12], although the treatment effect remains unclear. PSD is also associated with functional impairment, and rehabilitation is less effective; moreover, hospital stays are longer in PSD patients than in stroke patients without depression [13,14]. Some studies have reported that treatment for PSD also affects cognitive and functional improvements [4,12]. However, others have found no improvement in cognitive impairment, albeit significant reductions in depressive symptoms in stroke patients were noted [15,16]. Machine learning (ML) algorithms are widely used in the medical field for the prediction of disease diagnosis and treatments. Some recent studies reported that the ML method is more effective in predicting stroke outcomes than statistical methods or scoring systems [17,18,19]. In addition, various psychiatric disorders after stroke, including depression, were predictable using ML algorithms [20]. Some predictors for the treatment outcome of PSD were analyzed statistically in a previous study [21]; however, no studies have used an ML algorithm to predict PSD in stroke patients in combination with the treatment outcome of PSD based on comprehensive cognitive and functional analysis.

In this study, we aimed to use various ML algorithms to predict the occurrence and prognosis of PSD in stroke patients based on their cognitive and functional status and evaluated whether ML algorithms are superior to statistical methods.

## 2. Subjects and Methods

### 2.1. Subjects

A total of 623 patients who had a first-ever hemorrhagic or ischemic stroke were reviewed, and finally, 31 patients who were diagnosed with PSD on admission were included (Figure 1). PSD was confirmed by psychiatrists using the Diagnostic and Statistical Manual of Mental Disorders, fourth edition (DSM-IV) for major depressive disorder [22]. Any patients with any of the following conditions were excluded: preexisting major depressive disorder, dementia, Parkinson’s disease, any other brain lesions, no follow-up with a psychiatrists’ assessment for the progress of PSD, or no follow-up with a computerized neurocognitive test (CNT). Over the course of 4 weeks following their diagnosis, all PSD patients received psychiatric and medical treatments, including antidepressants (i.e., escitalopram, amitriptyline, or fluoxetine). A total of 35 age-matched patients who had a first-ever stroke without PSD were recruited as controls, and their cognitive and functional changes were compared with those of PSD patients.

### 2.2. Measurements

Basic characteristics including medical and family history, education period, and smoking habits were documented, and neurological, cognitive, and functional tests were performed on all subjects, including those in the control (*n* = 34) and PSD (*n* = 31) groups, at admission to a rehabilitation unit in the subacute stage following stroke and 4 weeks after initial assessments.

#### 2.2.1. Neurological Assessment

A neurological assessment scored on the National Institutes of Health Stroke Scale (NIHSS) [23] was performed. In addition, the type of stroke, which was classified as hemorrhagic or ischemic, and the laterality of stroke were evaluated by magnetic resonance imaging.

#### 2.2.2. Psychological Assessment

When subjects were admitted to a rehabilitation unit, the results of the Hamilton Rating Scale for Depression, a test commonly used to screen for and identify depression [24], were evaluated by a rehabilitation doctor, and the cutoff score was 10 [3]. Subsequently, a psychiatrist interviewed selected patients and diagnosed them with major depressive disorder according to the criteria for the DSM-IV (which specifies more than five different symptoms of depression from a list of nine, at least one of which was either depressed mood or loss of interest or pleasure for more than two weeks) [22,25]. All of the PSD patients were followed up 4 weeks later by the same psychiatrist to assess them for any improvements in their depressive symptoms. According to the psychiatrist’s follow-up results, PSD patients were divided into improved (Imp) and not improved (NoImp) groups (Figure 1).

#### 2.2.3. Cognitive Assessments

The Korean version of the Mini-Mental Status Examination (K-MMSE) and CNT were assessed in all subjects at admission to a rehabilitation unit, and participants were followed up 4 weeks after the initial assessment. The K-MMSE was modified from the original MMSE [26,27] by Kang et al. [28] and consists of 11 questions in the following five categories: orientation to place and time, registration, recall, attention and calculation, and language and complex commands; the total score ranges from 0 to 30. The CNT consists of 20 items in 4 major subtests: visual memory, language memory, visual perception, and language perception [29]. Each item is scored on a scale ranging from T-scores of 27 to 80. The total score ranges from 108 to 1600.

#### 2.2.4. Functional Assessments

The severity of impairment in ADL performance was evaluated using the Korean version of the Modified Barthel index (K-MBI) and functional independence measure (FIM) for all subjects at admission to a rehabilitation unit, and participants were followed up 4 weeks after the initial assessment. The K-MBI is composed of 10 items: hygiene (grooming), bathing, eating, toileting, stair-climbing, dressing, bowel control, bladder control, toilet transfer, and ambulation; scores range from 0 (completely dependent) to 100 (independent in basic ADL) [30,31]. The FIM is composed of 18 items and divided into six areas: self-care, sphincter control, transfer, locomotion, communication, and social. Scores range from 18 (completely dependent) to 126 (independent in basic ADL) [32].

### 2.3. ML Analysis

We used two sets of data: control vs. PSD and patients in the Imp vs. NoImp groups (Figure 2). For the process of feature selection, baseline characteristics and the initial cognitive and functional data were used initially. Input features were reduced repeatedly one by one, from the least important to the last, which is known as the wrapper method. This was done to select the best set of features for prediction based on feature importance; it turned out that performance was better when this approach was used. The prediction of PSD occurrence and prognosis was developed using 5 ML models of support vector machine linear (SVM_L), support vector machine with radial basis function (RBF) kernel (SVM_R), k-nearest neighbors (KNN), random forest (RF), and a voting ensemble (VE) algorithm from a retrospective study that included all subjects. Then, we assessed model performance using accuracy and receiver operating characteristic (ROC) curves. The average accuracy was improved from 53.21% to 60.61% in the SVM model by the wrapper method. The wrapper method was used to select the most important predictors, and data were analyzed by ML algorithms with 5- and 10-fold cross-validation. In 10-fold cross-validation, groups were randomly shuffled and partitioned into 10 groups, each of which was used as the test set, while the remaining nine were used for training. A total of 13 randomly selected test datasets were used for the prediction of PSD occurrence and prognosis. Decision tree classifiers are commonly used to provide a descriptive representation of a classifier. The inner nodes of the decision tree represent features, and the branches represent decision rules. In this paper, we used a decision tree classifier from the scikit-learn package with Gini impurity as a partitioning criterion [33].

### 2.4. Statistics

Statistical analyses were performed using PASW Statistics 18 for Windows (IBM Corp., New York, NY, USA). The Shapiro–Wilk test was performed to assess the normal distribution of all numerical data from each group. The likelihood ratio was obtained to analyze baseline categorical data, such as sex and the cause and laterality of stroke on initial assessment, among other aspects. The Mann–Whitney U test was used to compare numerical data of age, educational period, NIHSS score, and time since stroke between participants in the control and PSD groups and between participants with PSD who showed improvement in their symptoms and between participants with PSD without improvement in their symptoms. The Mann–Whitney U test was performed to compare the initial, follow-up, and gain values of the K-MMSE, CNT, K-MBI, and FIM total and subtest scores between participants in the control and PSD groups and between participants with PSD who showed improvement in their symptoms and participants with PSD without improvement in their symptoms. A Wilcoxon sum rank test was conducted to compare the initial values to the follow-up values of K-MMSE, CNT, FIM, and K-MBI total and subtest scores in the same subjects in the control and PSD groups. The Spearman rank correlation analysis was performed using statistically significant parameters that were determined by the Mann–Whitney U test between participants in the control and PSD groups and participants with PSD who showed improvement and participants with PSD without improvement in their symptoms. The logistic regression analysis was performed, and significant variables for the prediction of PSD (control vs. PSD groups) and PSD prognosis (Imp vs. NoImp) were determined using the forward Wald method. The predictive performance was considered based on the area under the ROC curve (AUC) with their 95% confidence intervals (CIs), sensitivity values, and specificity values. Statistical significance was set at *p* < 0.05.

## 3. Results

### 3.1. Baseline Characteristics

Among the 31 PSD patients, depression symptoms were improved in 13 patients (41.9%) 4 weeks after their initial assessment. The number of positive depressive symptoms among the nine-item depression module based on DSM-IV [34] showed no difference between NoImp and Imp groups initially (6.06 ± 1.39 and 6.69 ± 1.60 for NoImp and Imp groups, respectively), but the Imp group showed a smaller number of depressive symptoms at follow-up period (6.67 ± 1.53 and 2.08 ± 2.25 for NoImp and Imp groups, respectively, Table 1). We found that there were no differences in age, sex, educational period, onset from stroke to initial evaluation, type and laterality of stroke, NIHSS score, family history, mental disorder history, smoking year, history of diabetes mellitus or hypertension between participants in the control and PSD groups, and the educational period was different between participants in the NoImp and Imp groups (Table 1).

### 3.2. Cognitive and Functional Analysis Using Statistical Methods

Patients’ cognitive status was expressed as the total K-MMSE score (14.0 ± 8.4 and 14.1 ± 7.7 for participants in the control and PSD groups, respectively) and CNT score (419.1 ± 180.1 and 390.1 ± 127.8 for participants in the control and PSD groups, respectively) were not different between the groups initially (Table 2). Initial functional status based on the total scores of the K-MBI (25.7 ± 25.2 and 19.8 ± 15.5 for participants in the control and PSD groups, respectively) and FIM (46.3 ± 22.9 and 44.1 ± 15.5 for participants in the control and PSD groups, respectively) were also not different between participants in the control and PSD groups. During the 4-week follow-up period, the K-MMSE, K-MBI, and FIM total scores were improved for participants in both the control and PSD groups without differences between groups, but the CNT total score was not changed from the initial values (Table 2). The follow-up total K-MBI and FIM scores were lower for participants in the PSD group (40.1 ± 19.7 and 59.1 ± 17.5, respectively) than for participants in the control group (46.5 ± 28.6 and 64.7 ± 28.2, respectively). When comparing the PSD patients in the Imp group with those in the NoImp group, follow-up K-MMSE total scores were higher for patients in the Imp group (21.6 ± 6.8) than for those in the NoImp group (16.1 ± 8.2) (Table 2).

The detailed scores of the K-MMSE were divided into five categories. All categories except registration at the follow-up period showed improvement from initial values, but there was no difference between participants in the control and PSD groups (Table 3). When comparing participants with PSD in the Imp group with those in the NoImp group, the recall subscores of participants in the Imp group at the follow-up period (2.8 ± 2.2) were higher than those of participants in the NoImp group (0.9 ± 1.0) (Table 3).

Among the 12 subtests of the CNT, the initial visual attention omission error score of participants in the PSD group (30.5 ± 12.0) was lower than that of participants in the control group (38.2 ± 18.2), and the gain scores of auditory attention correct time standard deviation (SD) and visual attention commission error of participants in the PSD group (5.9 ± 12.9 and 5.4 ± 15.5, respectively) were different from those of participants in the control group (−7.5 ± 21.7 and 9.3 ± 13.9, respectively) (Table 4). The follow-up scores of most categories except auditory and visual attention omission error were improved for participants in the control group, whereas the follow-up scores of only four subtests, such as digit span backward (DSB) language memory, auditory attention correct response, and commission error, and visual attention correct response, improved for participants in the PSD group (Table 4).

Among the 10 items of the K-MBI, the initial subscore of dressing was significantly lower for participants in the PSD group (2.0 ± 1.9) than for participants in the control group (3.4 ± 2.7). In addition, follow-up subscores of bathing, toileting, stair-climbing, dressing, bladder control, transfer and ambulation, and gain subscores of bladder control were lower for participants in the PSD group than for participants in the control (Table 5). However, there was no difference in the initial, follow-up, or gain subscores of all items between participants with PSD in the Imp and NoImp groups (Table 5).

Among the six areas in the FIM, the follow-up subscores of self-care, transfer, and locomotion were significantly lower for participants in the PSD group (17.8 ± 7.3, 8.3 ± 3.9, and 4.3 ± 3.3, respectively) than for participants in the control group (23.2 ± 10.2, 12.2 ± 5.5 and 5.9 ± 3.7, respectively). In addition, all initial, follow-up, and gain subscores for participants in the Imp and NoImp groups were not significantly different (Table 6).

As a result of the correlation analysis between featured parameters, a strong correlation was observed between the total and subscores of the functional tests (K-MBI and FIM) in the control and PSD groups (Appendix A), and the total and subscores of the cognitive tests (K-MMSE and CNT) in the Imp and NoImp groups (Appendix A). Then, logistic regression analysis was performed, and two parameters, the initial subscore of auditory attention omission error on the CNT (CNT1_AA OE) and the initial subscore of bathing on the K-MBI (MBI1_Bat) for the comparison of participants in the control and PSD groups, and one parameter, educational period for the comparison of participants in the Imp and NoImp groups, were included. The AUC and accuracy used to classify the control and PSD groups were 0.706 and 0.696, respectively, and those used to classify the Imp and NoImp groups were 0.797 and 0.778, respectively (Table 7).

### 3.3. ML Analysis

The featured parameters were selected through a wrapper method, and each parameter was ranked by importance using support vector machine linear model-based recursive feature elimination (Table 8). The five featured parameters (MBI1_Amb, CNT1_AA OE, MMSE1_Rec, MBI1_Dre, FIM1_Loc) for the prediction of PSD (control vs. PSD groups) and five parameters (MBI1_Bla, MBI1_Bow, FIM1_Tra, CNT1_VM VSB, FIM1_Com) for the prediction of PSD prognosis (Imp vs. NoImp) were used for ML analysis.

When five ML algorithms (SVM_L, SVM_R, KNN, RF, and VE) with 5- or 10-fold cross-validation were compared, SVM_R with 10-fold cross-validation showed the best AUC for the prediction of PSD occurrence (0.711), and SVM_L with five-fold cross-validation showed the best AUC for the prediction of PSD prognosis (0.830) (Table 9). Accuracies of SVM_R with 10-fold cross-validation for the prediction of PSD occurrence and SVM_L with five-fold cross-validation for the prediction of PSD prognosis were 7.000 and 0.771, respectively, and which were analyzed using hyper-parameters (Appendix A). SVM_L with 10-fold cross-validation showed the best sensitivity (0.775) for the prediction of PSD occurrence and showed the best sensitivity (0.650) and specificity (0.950) for the prediction of PSD prognosis (Table 9).

The ROC curves of each ML algorithm and logistic regression analysis are shown in Figure 3. The mean AUC of SVM_R with 10-fold cross-validation for the prediction of PSD occurrence was 0.71 ± 0.12, which was comparable to that of logistic regression analysis (0.71 ± 0.07). The mean AUC of SVM_L with five-fold cross-validation for the prediction of PSD prognosis was also higher than that of logistic regression analysis (0.83 ± 0.14 vs. 0.80 ± 0.09) (Figure 3).

### 3.4. Decision-Making Model for the Prediction of PSD Occurrence and Prognosis

Decision tree classification models for the prediction of PSD occurrence were created using initial values of featured parameters, as shown in Table 8 (Figure 4). From this analysis, we can recursively construct a tree structure in which the input featured parameters and their values can be precisely assigned a given label by generating an appropriate partition and final decision of PSD for clinical use.

We found that the initial subscore of auditory attention omission error on the CNT was identified as the first single discriminator for group determination between participants in the control and PSD groups (Figure 4). The initial ADL and locomotor functions such as dressing and transfer on the K-MBI were also important discriminators to predict PSD occurrence (Figure 4).

## 4. Discussion

In this study, we found that numerous cognitive and functional statuses were associated with the occurrence of PSD in stroke patients (Table 5 and Table 6). The recall and auditory attention omission error among the subitems of the MMSE and CNT, and dressing and locomotor function including ambulation among the subitems of the K-MBI and FIM were considered to be important features to predict the occurrence of PSD ((A) in Table 8). From decision-making models, we found that initial scores of visual and auditory attention were important for the prediction of PSD occurrence. A previous study revealed that the severity of depression and the decrease in visual and auditory tension tend to correlate to a weak degree in PSD patients [35], and depressive patients also showed impaired visual attention omission errors in a meta-analysis [36].

PSD is frequently seen in stroke patients, and it might worsen their cognitive and functional recovery and quality of life [37]. The prevention of PSD has been suggested in previous studies using various nonpharmacological modalities and antidepressants [38]; however, the evidence is very limited, and more clinical trials are needed to confirm effective prevention methods for PSD [39]. There is no doubt that early detection and treatment of PSD can help improve a patient’s prognosis; therefore, it is important to identify modifiable risk factors and their application to stroke patients. From previous studies, major risk factors from meta-analysis still debated according to researchers were identified as follows: previous history of mental disorders, including depression or anxiety; diabetes mellitus; cognitive impairment and functional deficits, including impairment in ADL; and other factors, such as old age, female sex, lesion location, and stroke type [40,41]. It should be noted here that risk factors, which are determined by statistical methods, cannot be used to develop a predictive model directly because risk factors contain results, course, or complications of target diseases, as well as causative factors and can be used as an explanatory model [42].

ML methods are a useful tool to overcome the limitations of statistical methods and help us to find predictive models [43]. Most previous ML studies on stroke patients have focused on the prediction of stroke occurrence or outcome [44], and only one study has revealed the relationship between stroke and mood disorders, including depression, apathy, and anxiety, using ML analysis [20]. In this study, we used various ML algorithms to predict PSD occurrence and prognosis for the first time, and the SVM linear and SVM with an RBF kernel were optimal to develop a predictive model for PSD. The RBF kernel is commonly used in SVM classification, has the advantages of the KNN algorithm, and overcomes challenges of using the RBF alone, such as the space complexity problem [45]. We also tried to apply other ML algorithms, including KNNs and RFs, which were effective in predicting stroke occurrence and prognosis in previous studies [44]. The VE algorithm is the combination of all ML models that were used in this study to improve model performance; however, its AUC and accuracy were not higher than those of the SVM linear or SVM with the RBF kernel (Table 9). The VE model uses multiple models for the analysis, which might contain pros and cons of each model, and individual algorithms, which are superior to other algorithms, can show better performance.

Whereas cognitive impairment and PSD are highly connected with each other, the effect of PSD treatment on cognitive improvement was not clear in previous studies. In one study, improved PSD patients showed greater cognitive improvement than nonimproved PSD patients showed [46], but in another study, PSD treatment only helped to improve attention [47]. Previous studies also revealed that motor recovery is obvious in treated PSD patients [48]. In our study, cognitive impairments such as visual memory visual span backward of the CNT, were closely related to improvements in patients with PSD ((B) in Table 8); however, further studies are needed to further confirm these relationships. We found that the educational period was strongly associated with recovery from PSD in the logistic regression analysis (AUC = 0.80). In previous studies, educational level might be associated with anxiety, depression [49], and poststroke cognitive impairment [50] and be a protective factor against PSD [51,52].

The statistical method of logistic regression analysis did not show a higher AUC than the optimal ML algorithm for the prediction of PSD occurrence and prognosis (Figure 3). During statistical processing of logistic regression, only a few parameters were included for the comparison of participants in the control and PSD groups or the Imp and NoImp groups, whereas ML algorithms included five parameters for the detection of AUC and accuracy. The sample size was important to reduce bias in regression coefficients [53], and a smaller sample size might have influenced the low AUC and accuracy of the regression analysis in this study. Nevertheless, ML algorithms in this study showed comparable performance to the existing statistical methods and might have been a suitable method to overcome a small sample size.

The SVM linear and SVM with RBF showed the best AUC among various ML algorithms for the prediction of PSD occurrence and prognosis; however, the specificity for the prediction of PSD occurrence and the sensitivity for the prediction of PSD prognosis was low, which means that these models might not accurately detect PSD among stroke patients or PSD patients whose symptoms improved depending on the cut-off selected. We studied 65 stroke patients, including controls without PSD, for the development of ML algorithm-based prediction models of PSD occurrence and prognosis; notably, many previous studies could not evaluate more than 60 stroke patients [16,54,55]. Small sample sizes can cause some problems, such as generalization, in the field of ML. In the medical field, the small sample size is due to an imbalance in which the number of people with the disease is smaller than the number of people without the disease. This imbalance problem can be addressed by introducing a method of oversampling relatively small data [56]. In this study, we performed a 5- and 10-fold stratified cross-validation set to compensate for the small sample size [57]. Although there are differences according to the ML algorithms, no significant difference was found between the two methods for the prediction of PSD occurrence and prognosis (Table 9). A prospective study can provide more accurate clinical information; therefore, it is possible to develop better predictive models. Pertinently, some prospective studies have been designed to predict PSD, but they were not replicable in an independent stroke group [6].

In this study, we used various ML algorithms to predict PSD occurrence and prognosis for the first time and showed better performance than the statistical method. However, further studies with larger sample sizes and longer follow-up periods are needed to ascertain applications for clinical use.

## 5. Conclusions

We concluded that the occurrence and prognosis of PSD in stroke patients can be predicted effectively based on cognitive and functional status using ML algorithms.

## Figures and Tables

**Figure 1 jcm-11-02264-f001:**
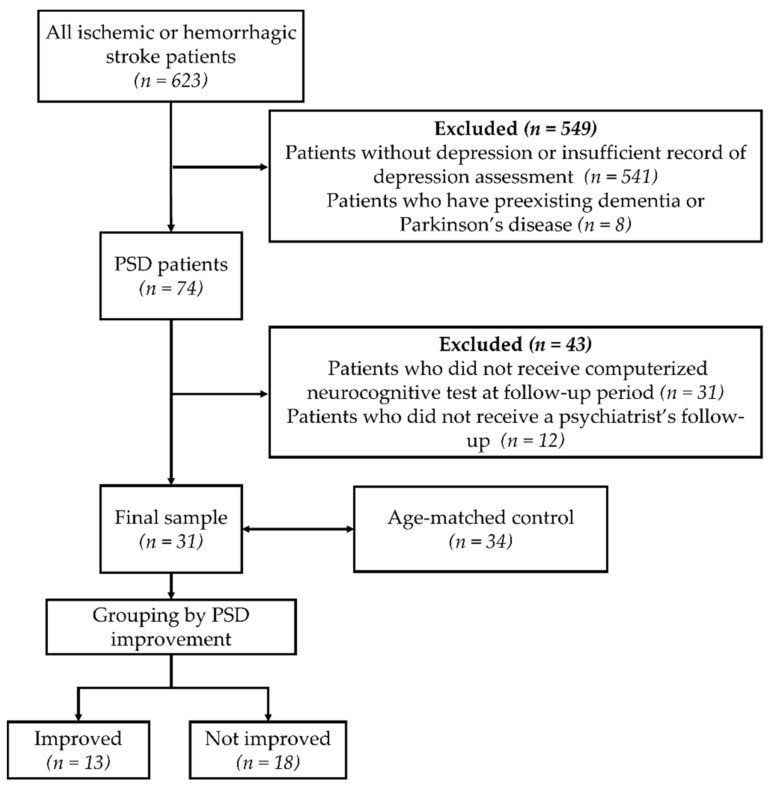
Flow and grouping of patients and controls. Abbreviation: PSD = poststroke depression.

**Figure 2 jcm-11-02264-f002:**
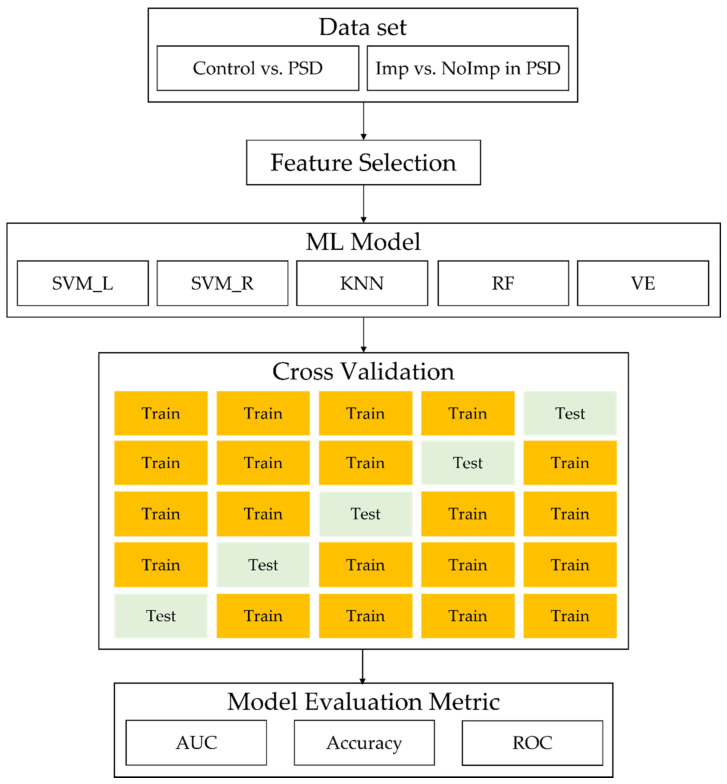
Flow of ML analysis. Abbreviations: PSD = poststroke depression; Imp = patients showing improvement in PSD symptoms; NoImp = patients showing no improvement in PSD symptoms; SVM_L = linear support vector machine; SVM_R = support vector machine with radial basis function kernel; KNN = k-nearest neighbors; RF = random forest; VE = voting ensemble; AUC = area under the curve; ROC = receiver operating characteristic.

**Figure 3 jcm-11-02264-f003:**
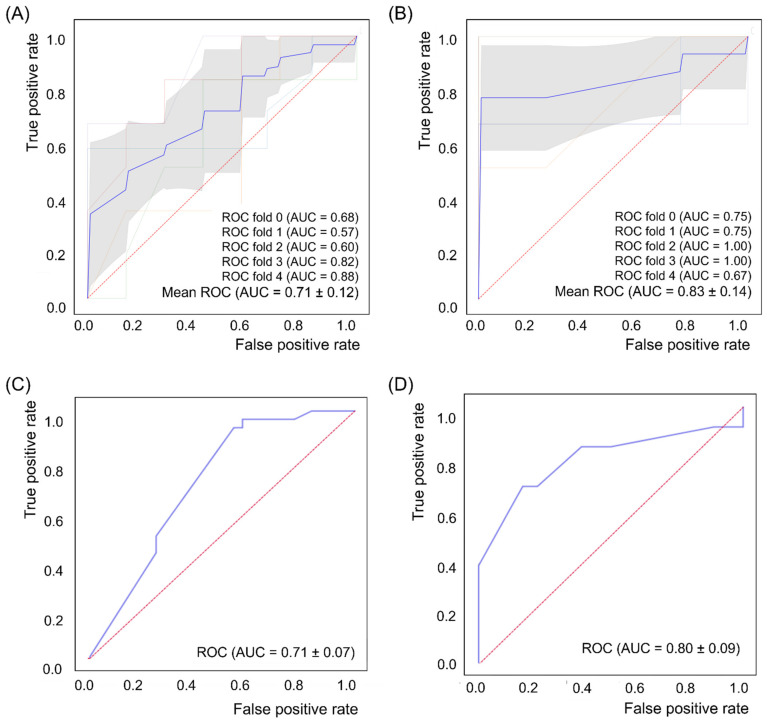
Receiver operating characteristic (ROC) curve for support vector machine with radial basis function kernel function algorithm with 10-fold cross-validation in the prediction of poststroke depression (PSD) occurrence (control vs. PSD, (**A**)), support vector machine linear algorithm with five-fold cross-validation in the prediction of PSD prognosis (improved vs. not improved, (**B**)), and ROC curve for logistic regression analysis in the prediction of PSD occurrence (**C**) and PSD prognosis (**D**).

**Figure 4 jcm-11-02264-f004:**
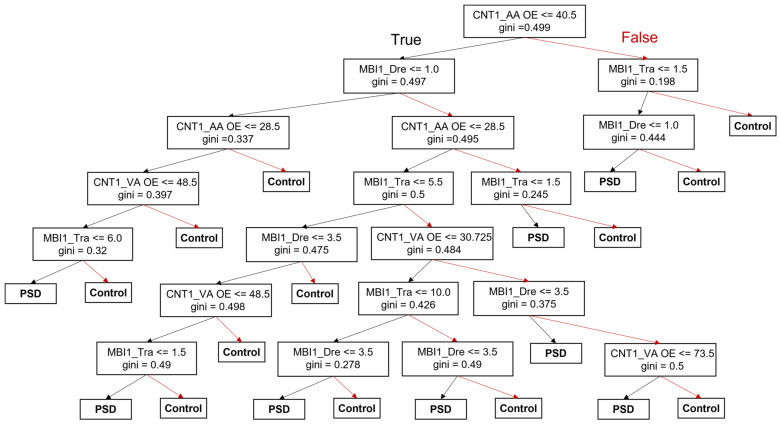
Decision tree models to classify controls and patients with PSD. Abbreviations: PSD = poststroke depression; CNT1_AA OE = Initial subscore of auditory attention omission error on the CNT; CNT1_VA OE = Initial subscore of visual attention omission error on the CNT; MBI1_Tra = Initial subscore of transfer on the K-MBI; MBI1_Dre = Initial subscore of dressing on the K-MBI. Black arrows = true, red arrows = false.

**Table 1 jcm-11-02264-t001:** Baseline characteristics of controls and patients with PSD.

	Controls	PSD Patients				*p* Value ^2^
		NoImp	Imp	*p* Value ^1^	All	
No. of subjects	34	18	13	0.798	31	
Age (years)	64.6 ± 15.9	62.3 ± 11.5	63.5 ± 13.9	0.779	62.8 ± 15.9	0.563
Female, no. (%)	15 (44)	7 (39)	8 (61)	0.285	21 (58)	0.546
Educational period (years)	9.2 ± 5.0	7.0 ± 3.4	9.8 ± 4.0	0.027 *	8.2 ± 3.9	0.406
Onset (days)	74.3 ± 68.9	71.2 ± 41.3	69.5 ± 25.4	0.890	70.5 ± 35.0	0.250
No. of depressive symptoms						
Initial	-	6.06 ± 1.39	6.69 ± 1.60	0.260	6.32 ± 1.49	-
Follow-up	-	6.67 ± 1.53	2.08 ± 2.25	0.000 *	4.74 ± 2.94	-
Type of stroke, no. (%)						
Hemorrhagic stroke	15 (44)	3 (23)	9 (50)	0.833	12 (39)	0.661
Ischemic stroke	19 (56)	10 (77)	9 (50)	0.833	19 (61)	0.661
Side of hemiplegia (Rt:Lt:both)	14 (42):12 (35):8 (23)	6 (33):7 (39):5 (28)	3 (23):8 (61):2 (16)	0.462	9 (29):15 (48):7 (23)	0.518
Family Hx. (medical disorders)	12 (35.3)	6 (33)	6 (46)	0.470	12 (39)	0.777
Family Hx. (mental disorders)	0 (0)	1 (5)	0 (0)	0.388	1 (3)	0.295
Smoking (years)	8.2 ± 15.8	6.1 ± 14.4	9.6 ± 17.1	0.567	7.5 ± 15.4	0.880
Diabetes mellitus	8 (23)	4 (22)	6 (46)	0.160	10 (30)	0.436
Hypertension	10 (56)	10 (55)	6 (46)	0.605	16 (52)	0.288
NIHSS score	7.6 ± 5.6	7.3 ± 7.0	8.0 ± 4.3	0.535	6.8 ± 3.4	0.826

Note: Values are presented as the number of subjects (%) or mean ± standard deviation. ^1^ *p* values between participants in the NoImp and Imp groups determined by the Mann–Whitney U test or the likelihood ratio, ^2^ *p* values between participants in the control and PSD groups determined by the Mann–Whitney U test or the likelihood ratio, * *p* < 0.05. Abbreviations: PSD = poststroke depression; NoImp = PSD patients with no symptom improvement; Imp = PSD patients with improvement in their symptoms; NIHSS = National Institutes of Health Stroke Scale.

**Table 2 jcm-11-02264-t002:** Initial and follow-up results of all cognitive and functional tests for controls and PSD patients.

	Controls	PSD Patients				*p* Value ^2^
		NoImp	Imp	*p* Value ^1^	All	
K-MMSE						
Initial	14.0 ± 8.4	12.4 ± 7.6	16.3 ± 7.5	0.206	14.1 ± 7.7	0.787
Follow-up	19.3 ± 9.3	16.1 ± 8.2	21.6 ± 6.8	0.037 *	18.7 ± 8.0	0.491
Gain	5.4 ± 6.8	4.3 ± 3.2	5.4 ± 3.1	0.395	4.7 ± 3.1	0.535
*p* value ^3^	0.000 *	0.001 *	0.000 *		0.000 *	
CNT						
Initial	419.1 ± 180.2	371.5 ± 114.4	415.8 ± 145.1	0.603	390.1 ± 127.8	0.427
Follow-up	449.5 ± 220.7	422.9 ± 191.7	492.3 ± 230.0	0.526	449.9 ± 203.7	0.980
Gain	33.6 ± 70.9	49.1 ± 64.5	36.0 ± 57.7	0.618	44.0 ± 60.5	0.580
*p* value ^3^	0.119	0.241	0.237		0.088	
K-MBI						
Initial	25.7 ± 25.2	19.8 ± 15.5	25.7 ± 25.2	0.718	19.8 ± 15.5	0.103
Follow-up	46.5 ± 28.6	40.1 ± 19.7	46.5 ± 28.6	0.330	40.1 ± 19.7	0.006 *
Gain	18.7 ± 12.7	20.2 ± 15.5	18.7 ± 12.7	0.703	20.2 ± 15.5	0.276
*p* value ^3^	0.000 *	0.001 *	0.002 *		0.000 *	
FIM						
Initial	46.3 ± 22.9	44.1± 15.5	46.3 ± 22.9	0.904	44.1 ± 15.5	0.285
Follow-up	64.7 ± 28.2	59.1 ± 17.5	64.7 ± 28.2	0.525	59.1 ± 17.5	0.013 *
Gain	17.0 ± 11.9	16.5 ± 8.2	17.0 ± 11.9	0.925	16.5 ± 8.2	0.132
*p* value ^3^	0.000 *	0.000 *	0.005 *		0.000 *	

Note: Values are presented as the mean ± standard deviation. ^1^ *p* values between participants in the NoImp and Imp groups determined by the Mann–Whitney U test, ^2^ *p* values between participants in the control and PSD groups determined by the Mann–Whitney U test, ^3^ *p* values between initial and follow-up values determined by the Wilcoxon sum rank test, * *p* < 0.05. Abbreviations: PSD = poststroke depression; NoImp = PSD patients showing no symptom improvement; Imp = PSD patients showing improvement in their symptoms; K-MMSE = Korean version of the Mini-Mental Status Examination; CNT = computerized neurocognitive test; K-MBI = Korean version of the modified Barthel index; FIM = functional independence measure.

**Table 3 jcm-11-02264-t003:** Initial and follow-up results of K-MMSE categories for controls and PSD patients.

	Controls	PSD Patients				*p* Value ^2^
		NoImp	Imp	*p* Value ^1^	All	
Orientation						
Initial	5.1 ± 3.1	4.2 ± 3.5	5.5 ± 3.4	0.224	4.7 ± 3.5	0.630
Follow-up	6.8 ± 3.4	5.3 ± 3.3	7.5 ± 2.6	0.143	6.2 ± 3.2	0.355
Gain	1.7 ± 2.8	1.4 ± 2.1	1.4 ± 2.2	0.688	1.4 ± 2.1	0.916
*p* value ^3^	0.003 *	0.020 *	0.077		0.003 *	
Registration						
Initial	2.3 ± 1.1	2.3 ± 1.2	2.9 ± 0.3	0.138	2.6 ± 1.0	0.254
Follow-up	2.5 ± 1.2	2.6 ± 1.4	3.0 ± 0.0	0.701	2.8 ± 1.1	0.200
Gain	0.2 ± 0.9	0.4 ± 1.0	0.1 ± 0.3	0.544	0.3 ± 0.8	0.983
*p* value ^3^	0.196	0.131	0.343		0.084	
Recall						
Initial	0.9 ± 1.8	0.5 ± 0.6	1.2 ± 1.5	0.364	0.8 ± 1.1	0.446
Follow-up	1.9 ± 2.1	0.9 ± 1.0	2.8 ± 2.2	0.027 *	1.6 ± 1.8	0.894
Gain	0.9 ± 1.5	0.4 ± 0.7	1.6 ± 1.6	0.026 *	0.9 ± 1.2	0.601
*p* value ^3^	0.003 *	0.038 *	0.011 *		0.002 *	
Attention and calculation						
Initial	1.1 ± 1.3	0.5 ± 0.8	1.2 ± 1.2	0.063	0.8 ± 1.0	0.495
Follow-up	1.7 ± 1.3	1.4 ± 1.4	2.0 ± 0.9	0.374	1.7 ± 1.2	0.946
Gain	0.5 ± 1.3	0.9 ± 1.0	0.7 ± 1.7	0.695	0.8 ± 1.3	0.439
*p* value ^3^	0.013 *	0.010 *	0.226		0.007 *	
Language and complex commands						
Initial	4.6 ± 2.7	4.9 ± 2.7	5.5 ± 2.6	0.762	5.2 ± 2.6	0.341
Follow-up	6.3 ± 3.0	5.8 ± 2.8	7.4 ± 1.5	0.080	6.4 ± 2.5	0.696
Gain	1.4 ± 2.2	1.1 ± 1.1	1.6 ± 1.8	0.571	1.3 ± 1.4	0.659
*p* value ^3^	0.000 *	0.003 *	0.022 *		0.000 *	

Note: Values are presented as the mean ± standard deviation. ^1^ *p* values between participants in the NoImp and Imp groups determined by the Mann–Whitney U test, ^2^ *p* values between participants in the control and PSD groups determined by the Mann–Whitney U test, ^3^ *p* values between initial and follow-up values determined by the Wilcoxon sum rank test, * *p* < 0.05. Abbreviations: PSD = poststroke depression; NoImp = PSD patients showing no symptom improvement; Imp = PSD patients showing improvement in their symptoms.

**Table 4 jcm-11-02264-t004:** Initial and follow-up results of subtests of the CNT for controls and PSD patients.

	Controls	PSD Patients				*p* Value ^2^
		NoImp	Imp	*p* Value ^1^	All	
Language memory digit span forward						
Initial	30.4 ± 5.1	29.2 ± 4.2	32.2 ± 7.3	0.090	30.9 ± 5.9	0.869
Follow-up	31.5 ± 6.9	29.0 ± 3.2	35.1 ± 5.2	0.017 *	31.5 ± 6.9	0.592
Gain	2.3 ± 3.3	0.5 ± 2.2	1.4 ± 5.2	0.728	0.8 ± 3.3	0.105
*p* value ^3^	0.005 *	0.465	1.00		0.483	
Language memory digit span backward						
Initial	29.9 ± 5.6	28.3 ± 2.5	30.4 ± 7.4	0.689	29.2 ± 5.1	0.934
Follow-up	33.0 ± 11.2	29.0 ± 4.8	35.6 ± 8.2	0.013 *	32.0 ± 7.8	1.000
Gain	3.9 ± 8.4	1.1 ± 2.8	7.0 ± 8.3	0.090	3.4 ± 6.1	0.847
*p* value ^3^	0.005 *	0.180	0.046 *		0.017 *	
Visual memory visual span forward						
Initial	30.7 ± 7.5	28.9 ± 3.0	33.5 ± 9.3	0.201	30.8 ± 6.7	0.820
Follow-up	32.0 ± 7.8	29.6 ± 3.4	32.6 ± 7.6	0.496	30.8 ± 5.4	0.684
Gain	2.7 ± 6.3	1.1 ± 2.9	−0.9 ± 5.6	0.619	0.3 ± 6.3	0.373
*p* value ^3^	0.016 *	0.225	1.00		0.440	
Visual memory visual span backward						
Initial	30.6 ± 6.1	28.8 ± 3.0	32.9 ± 8.0	0.215	30.5 ± 5.9	0.650
Follow-up	32.0 ± 5.9	29.8 ± 4.3	34.7 ± 10.6	0.230	31.7 ± 7.5	0.593
Gain	2.5 ± 4.6	0.6 ± 4.8	1.4 ± 5.7	0.814	0.9 ± 5.0	0.424
*p* value ^3^	0.012 *	0.439	0.465		0.323	
Auditory attention correct response						
Initial	29.8 ± 6.6	27.8 ± 3.2	29.0 ± 3.2	0.399	28.3 ± 4.1	0.320
Follow-up	31.9 ± 11.2	31.9 ± 7.0	36.1 ± 18.3	1.000	33.3 ± 12.3	0.361
Gain	3.7 ± 7.7	3.6 ± 5.0	6.7 ± 12.1	0.826	4.9 ± 8.4	0.465
*p* value ^3^	0.027 *	0.068	0.102		0.027 *	
Auditory attention commission error						
Initial	29.8 ± 6.6	27.8 ± 3.2	29.0 ± 3.2	0.399	28.3 ± 4.1	0.320
Follow-up	31.9 ± 11.2	31.9 ± 7.0	36.1 ± 18.3	1.000	33.6 ± 12.6	0.361
Gain	3.7 ± 7.7	3.6 ± 5.0	6.7 ± 12.1	0.826	4.9 ± 8.4	0.465
*p* value ^3^	0.027 *	0.068	0.102		0.027 *	
Auditory attention omission error						
Initial	38.1 ± 17.2	34.5 ± 14.8	33.6 ± 13.7	0.723	28.7 ± 5.7	0.481
Follow-up	36.0 ± 15.1	41.3 ± 21.3	36.3 ± 18.5	0.711	31.2 ± 9.4	0.892
Gain	−5.9 ± 22.5	5.5 ± 10.5	5.7 ± 16.0	0.585	1.4 ± 10.7	0.052
*p* value ^3^	0.221	0.109	0.414		0.075	
Auditory attention correct time SD						
Initial	38.1 ± 17.2	34.5 ± 14.8	33.6 ± 13.7	0.773	34.1 ± 14.1	0.232
Follow-up	36.1 ± 15.1	41.3 ± 21.3	36.3 ± 18.5	0.740	39.2 ± 19.8	0.924
Gain	−7.5 ± 21.7	6.1 ± 10.9	5.7 ± 16.0	0.521	5.9 ± 12.9	0.026 *
*p* value ^3^	0.026 *	0.068	0.414		0.085	
Visual attention correct response						
Initial	33.2 ± 12.7	30.1 ± 9.9	35.4 ± 11.9	0.308	31.4 ± 11.1	0.617
Follow-up	38.7 ± 14.5	37.7 ± 19.2	34.1 ± 16.0	0.669	36.2 ± 17.5	0.202
Gain	9.0 ± 13.2	11.6 ± 19.2	−2.4 ± 5.8	0.041 *	5.4 ± 16.2	0.174
*p* value ^3^	0.006 *	0.078	0.197		0.026 *	
Visual attention commission error						
Initial	33.4 ± 13.1	30.1 ± 10.7	32.4 ± 12.0	0.381	31.1 ± 11.1	0.223
Follow-up	38.7 ± 14.5	37.7 ± 19.2	34.1 ± 16.0	0.606	36.2 ± 17.5	0.072
Gain	9.3 ± 13.9	10.1 ± 18.7	−1.3 ± 5.3	0.138	5.4 ± 15.5	0.049 *
*p* value ^3^	0.006 *	0.078	0.414		0.288	
Visual attention omission error						
Initial	38.2 ± 18.2	30.1 ± 12.5	30.9 ± 11.9	0.481	30.5 ± 12.0	0.027 *
Follow-up	40.4 ± 20.5	31.0 ± 6.8	33.1 ± 16.3	0.396	31.9 ± 11.2	0.303
Gain	1.0 ± 20.7	4.0 ± 6.8	−1.1 ± 22.6	0.253	1.9 ± 15.0	0.677
*p* value ^3^	0.671	0.066	0.655		0.344	
Visual attention correct time SD						
Initial	29.6 ± 5.8	29.3 ± 4.1	30.6 ± 6.4	0.651	29.9 ± 5.2	0.640
Follow-up	30.3 ± 4.6	29.5 ± 3.1	31.6 ± 6.3	0.622	30.5 ± 4.8	0.885
Gain	3.8 ± 7.7	0.4 ± 6.3	1.3 ± 2.9	0.763	0.8 ± 4.9	0.438
*p* value ^3^	0.010 *	0.468	0.257		0.205	

Note: Values are presented as the mean ± standard deviation. ^1^ *p* values between participants in the NoImp and Imp groups determined by the Mann–Whitney U test, ^2^
*p* values between participants in the control and PSD groups determined by the Mann–Whitney U test, ^3^ *p* values between initial and follow-up values determined by the Wilcoxon sum rank test, * *p* < 0.05. Abbreviations: PSD = poststroke depression; NoImp = PSD patients showing no symptom improvement; Imp = PSD patients showing improvement in their symptoms; SD = standard deviation.

**Table 5 jcm-11-02264-t005:** Initial and follow-up results of subtests of the K-MBI for controls and PSD patients.

	Controls	PSD Patients				*p* Value ^2^
		NoImp	Imp	*p* Value ^1^	All	
Hygiene						
Initial	2.1 ± 1.8	1.8 ± 1.5	1.6 ± 2.0	0.507	1.7 ± 1.7	0.369
Follow-up	3.3 ± 1.6	2.6 ± 1.5	3.2 ± 1.3	0.414	2.9 ± 1.4	0.166
Gain	1.2 ± 1.7	0.9 ± 1.5	1.6 ± 1.7	0.277	1.2 ± 1.6	0.628
*p* value ^3^	0.001 *	0.027 *	0.017 *		0.001 *	
Bathing						
Initial	1.1 ± 1.2	0.6 ± 0.5	0.3 ± 0.5	0.101	0.5 ± 0.5	0.075
Follow-up	2.4 ± 1.6	1.4 ± 1.2	1.7 ± 1.5	0.567	1.5 ± 1.3	0.030 *
Gain	1.4 ± 1.6	0.7 ± 1.0	1.4 ± 1.4	0.164	1.0 ± 1.2	0.324
*p* value ^3^	0.000 *	0.015 *	0.016 *		0.001 *	
Eating						
Initial	3.7 ± 3.5	3.0 ± 3.0	3.3 ± 3.9	0.950	3.1 ± 3.3	0.543
Follow-up	6.6 ± 2.9	5.2 ± 3.1	5.7 ± 3.4	0.662	5.4 ± 3.2	0.146
Gain	2.9 ± 3.4	2.1 ± 2.6	2.5 ± 3.0	0.658	2.3 ± 2.7	0.531
*p* value ^3^	0.000 *	0.011 *	0.024 *		0.001 *	
Toileting						
Initial	2.9 ± 3.5	1.2 ± 2.2	2.3 ± 3.5	0.526	1.6 ± 2.8	0.097
Follow-up	5.5 ± 3.7	3.4 ± 2.8	4.3 ± 3.8	0.679	3.8 ± 3.2	0.047 *
Gain	2.8 ± 3.9	2.6 ± 2.8	2.0 ± 3.1	0.455	2.4 ± 2.9	0.773
*p* value ^3^	0.001 *	0.006 *	0.058		0.001 *	
Stair-climbing						
Initial	0.3 ± 1.4	0.0 ± 0.0	0.4 ± 1.4	0.239	0.2 ± 0.9	0.613
Follow-up	3.2 ± 3.7	0.3 ± 1.2	1.5 ± 3.2	0.273	0.8 ± 2.3	0.002 *
Gain	2.9 ± 3.6	0.3 ± 1.2	1.0 ± 2.5	0.314	0.6 ± 1.8	0.003 *
*p* value ^3^	0.001 *	0.317	0.180		0.109	
Dressing						
Initial	3.4 ± 2.7	2.2 ± 1.7	1.6 ± 2.1	0.230	2.0 ± 1.9	0.030 *
Follow-up	6.1 ± 3.2	3.9 ± 2.5	4.9 ± 3.1	0.453	4.3 ± 2.8	0.029 *
Gain	2.8 ± 3.2	1.9 ± 2.1	3.0 ± 3.3	0.388	2.3 ± 2.6	0.694
*p* value ^3^	0.000 *	0.008 *	0.028 *		0.001 *	
Bowel control						
Initial	5.3 ± 4.4	4.6 ± 4.5	4.8 ± 5.1	0.831	4.7 ± 4.7	0.448
Follow-up	8.4 ± 3.3	7.4 ± 3.7	6.7 ± 4.2	0.734	7.1 ± 3.9	0.190
Gain	3.2 ± 4.2	3.1 ± 4.1	2.0 ± 3.2	0.558	2.6 ± 3.8	0.921
*p* value ^3^	0.001 *	0.013 *	0.039 *		0.002 *	
Bladder control						
Initial	4.8 ± 4.5	3.1 ± 4.1	4.9 ± 4.9	0.169	3.9 ± 4.5	0.291
Follow-up	8.2 ± 3.4	5.8 ± 4.3	6.5 ± 4.5	0.548	6.1 ± 4.3	0.047 *
Gain	3.4 ± 4.3	2.5 ± 4.2	1.6 ± 2.8	0.724	2.1 ± 3.6	0.314
*p* value ^3^	0.000 *	0.031 *	0.066		0.007 *	
Transfer						
Initial	6.2 ± 5.0	3.0 ± 3.7	5.4 ± 5.9	0.408	4.0 ± 4.8	0.050
Follow-up	10.1 ± 4.4	6.9 ± 4.0	7.5 ± 5.7	0.697	7.1 ± 4.6	0.013 *
Gain	4.1 ± 4.5	4.5 ± 4.1	2.2 ± 3.5	0.137	3.6 ± 4.0	0.630
*p* value ^3^	0.000 *	0.003 *	0.058		0.001 *	
Ambulation						
Initial	2.4 ± 4.1	0.3 ± 0.5	1.1 ± 2.4	0.669	0.6 ± 1.6	0.058
Follow-up	6.8 ± 5.7	2.6 ± 3.5	3.4 ± 4.9	0.770	2.9 ± 4.0	0.009 *
Gain	4.5 ± 5.0	2.4 ± 3.4	2.1 ± 3.2	0.749	2.3 ± 3.3	0.129
*p* value ^3^	0.000 *	0.006 *	0.042 *		0.001 *	

Note: Values are presented as the mean ± standard deviation. ^1^ *p* values between participants in the NoImp and Imp groups determined by the Mann–Whitney U test, ^2^ *p* values between participants in the control and PSD groups determined by the Mann–Whitney U test, ^3^ *p* values between initial and follow-up values determined by the Wilcoxon sum rank test, * *p* < 0.05. Abbreviations: PSD = poststroke depression; NoImp = PSD patients showing no symptom improvement; Imp = PSD patients showing improvement in their symptoms.

**Table 6 jcm-11-02264-t006:** Initial and follow-up results of the FIM subtests for controls and PSD patients.

	Controls	PSD Patients				*p* Value ^2^
		NoImp	Imp	*p* Value ^1^	All	
Self-care						
Initial	15.4 ± 8.0	12.4 ± 5.0	11.6 ± 6.0	0.468	12.1 ± 5.4	0.102
Follow-up	23.2 ± 10.2	17.0 ± 6.2	19.0 ± 8.8	0.588	17.8 ± 7.3	0.033 *
Gain	7.8 ± 8.8	4.9 ± 3.9	6.5 ± 6.0	0.759	5.5 ± 4.8	0.783
*p* value ^3^	0.000 *	0.000 *	0.005 *		0.000 *	
Sphincter control						
Initial	7.8 ± 5.1	6.1 ± 4.2	7.6 ± 5.7	0.606	6.7 ± 4.9	0.350
Follow-up	11.4 ± 4.2	9.9 ± 4.2	9.6 ± 5.3	0.921	9.8 ± 4.6	0.200
Gain	3.6 ± 4.4	4.0 ± 4.0	2.0 ± 2.9	0.090	3.2 ± 3.7	0.847
*p* value ^3^	0.000 *	0.001 *	0.017 *		0.000 *	
Transfer						
Initial	8.3 ± 5.2	5.2 ± 3.1	7.2 ± 4.5	0.296	6.0 ± 3.8	0.052
Follow-up	12.2 ± 5.5	7.9 ± 2.5	8.9 ± 5.4	0.757	8.3 ± 3.9	0.004 *
Gain	3.9 ± 4.4	3.3 ± 2.2	1.7 ± 2.8	0.110	2.7 ± 2.5	0.530
*p* value ^3^	0.000 *	0.001 *	0.074		0.000 *	
Locomotion						
Initial	3.2 ± 2.5	2.3 ± 0.8	2.8 ± 1.5	0.497	2.5 ± 1.1	0.231
Follow-up	5.9 ± 3.7	4.1 ± 2.8	4.6 ± 4.2	0.541	4.3 ± 3.3	0.032 *
Gain	2.7 ± 3.2	1.8 ± 2.8	1.7 ± 2.9	0.315	1.8 ± 2.8	0.159
*p* value ^3^	0.000 *	0.012 *	0.066		0.002 *	
Communication						
Initial	7.9 ± 3.3	8.1 ± 3.5	7.7 ± 3.9	0.777	7.9 ± 3.6	0.925
Follow-up	10.2 ± 3.4	8.8 ± 3.0	9.6 ± 3.4	0.668	9.1 ± 3.1	0.169
Gain	2.3 ± 2.3	1.0 ± 1.8	1.9 ± 2.0	0.194	1.4 ± 1.9	,138
*p* value ^3^	0.000 *	0.034 *	0.016 *		0.001 *	
Social						
Initial	10.2 ± 5.1	10.1 ± 4.7	9.2 ± 5.3	0.627	9.7 ± 4.9	0.808
Follow-up	13.7 ± 5.4	11.5 ± 4.8	13.0 ± 5.5	0.654	12.1 ± 5.1	0.177
Gain	3.5 ± 4.1	1.6 ± 2.7	3.3 ± 2.8	0.152	2.3 ± 2.8	0.370
*p* value ^3^	0.000 *	0.000 *	0.000 *		0.000 *	

Note: Values are presented as the mean ± standard deviation. ^1^ *p* values between participants in the NoImp and Imp groups determined by the Mann–Whitney U test, ^2^ *p* values between participants in the control and PSD groups determined by the Mann–Whitney U test, ^3^ *p* values between initial and follow-up values determined by the Wilcoxon sum rank test, * *p* < 0.05. Abbreviations: PSD = poststroke depression; NoImp = PSD patients showing no symptom improvement; Imp = PSD patients showing improvement in their symptoms.

**Table 7 jcm-11-02264-t007:** Parameters of the logistic regression and their assessment.

Classification	F-Value	*p*-Value	Effect Size		AUC	Accuracy	Sensitivity	Specificity
Control vs. PSD	12.975	0.002	CNT1_AA OE	0.933	0.706	0.696	0.967	0.419
			MBI1_Bat	0.426				
Imp vs. NoImp	13.296	0.001	Edu_Per	1.616	0.797	0.778	0.692	0.833

Abbreviations: PSD = poststroke depression; Imp = PSD patients showing improvement in their symptoms; NoImp = PSD patients showing no symptom improvement; AUC = area under the curve; CNT1_AA OE = the initial subscore of auditory attention omission error on the CNT; MB1_Bat = the initial subscore of bathing on the K-MBI; Edu_Per = educational period.

**Table 8 jcm-11-02264-t008:** Lists of feature selection for the prediction of PSD (control vs. PSD, A) and the prognosis of PSD (Imp vs. NoImp, B).

Rank	Parameter Name
**(A) Control vs. PSD groups**	
1	Initial subscore of ambulation on the K-MBI (MBI1_Amb)
2	Initial subscore of auditory attention omission error on the CNT (CNT1_AA OE)
3	Initial subscore of recall on the K-MMSE (MMSE1_Rec)
4	Initial subscore of dressing on the K-MBI (MBI1_Dre)
5	Initial subscore of locomotion on the FIM (FIM1_Loc)
**(B) Imp vs. NoImp groups**	
1	Initial subscore of bladder control on the K-MBI (MBI1_Bla)
2	Initial subscore of bowel control on the K-MBI (MBI1_Bow)
3	Initial subscore of transfer on the FIM (FIM1_Tra)
4	Initial subscore of visual memory visual span backward on the CNT (CNT1_VM VSB)
5	Initial subscore of communication on the FIM (FIM1_Com)

Abbreviations: PSD = poststroke depression; Imp = PSD patients showing improvement in their symptoms; NoImp = PSD patients showing no symptom improvement.

**Table 9 jcm-11-02264-t009:** AUC, accuracy, sensitivity, and specificity of each ML algorithm.

ML Models	5-Fold Cross-Validation				10-Fold Cross-Validation			
	AUC	Accuracy	Sensitivity	Specificity	AUC	Accuracy	Sensitivity	Specificity
**Control vs. PSD**								
SVM_L	0.690	0.600	0.748	0.557	0.706	0.636	0.775	0.542
SVM_R	0.708	0.646	0.681	0.495	0.711	0.700	0.742	0.517
KNN	0.659	0.538	0.743	0.352	0.681	0.579	0.742	0.425
RF	0.685	0.538	0.619	0.557	0.696	0.560	0.767	0.600
VE	0.675	0.615	0.676	0.552	0.646	0.650	0.708	0.517
**Imp vs. NoImp**								
SVM_L	0.830	0.771	0.600	0.883	0.797	0.775	0.650	0.950
SVM_R	0.496	0.648	0.267	0.817	0.722	0.708	0.300	0.800
KNN	0.635	0.681	0.300	0.950	0.674	0.742	0.300	0.850
RF	0.760	0.743	0.467	0.867	0.624	0.717	0.500	0.950
VE	0.784	0.743	0.533	0.867	0.747	0.733	0.450	0.90

Abbreviations: PSD = poststroke depression; Imp = PSD patients showing improvement in their symptoms; NoImp = PSD patients showing no symptom improvement; SVM_L = linear support vector machine; SVM_R = support vector machine with radial basis function kernel function; KNN = k-nearest neighbors; RF = random forest; VE = voting ensemble; AUC = area under the curve.

## Data Availability

The data presented in this study are available from the corresponding authors upon reasonable request.

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
