# Peer review of "Prediction of Poststroke Depression Based on the Outcomes of Machine Learning Algorithms"

_jcm, 2022, doi:10.3390/jcm11082264_

Round 1

Reviewer 1 Report

This study attempts to predict prognosis and occurrence of post-stroke depression using regression and machine learning approaches. While interesting, there are various conceptual and methodological shortcomings in this study,Please see my details comments below.

Post-stroke depression typically develops 3-6 months after stroke. Depressive symptoms in the acute/ subacute phase are often found to subside and are possibly due to circumstances surrounding adjustment to daily activities/ functions as a result of stroke. Why were patients required to have PSD on admission? It would be much more valuable to use all 623 patients with stroke and use machine learning to predict who develops PSD 3-6 months later, as this is the more persistent and severe type of depression, that hinders further recovery and return to daily living activities.

In Figure 1, “Final population” should be renamed to “final sample”

Please explain why a cut-off score of 10 on the HAMD was chosen. Is this based on validity and reliability measures in stroke samples? If so, please reference the studies.

Selecting variables based on significant correlations is a biased approach. Because it is already known that the variables are associated, one knows in advance that the regression is likely to be significant. Please also provide the overall statistics, (F; p; effect size) for the logistic regression.

“The featured parameters were selected through statistical analysis” again, does this mean that all variables with significant correlations were included? This is not an appropriate way to conduct regression or ML analyses. All variables should at least initially be included in the model.

The majority of variables included in the model are scores collected at follow-up. However, the aim of this study is to PREDICT the occurrence and prognosis of PSD. In order to do that and have any clinically meaningful relevance, only values from the initial assessment should be used to PREDICT outcome at the second assessment. It is likely that patients without PSD at follow-up differ on so many measures because they overall recovered better. By including these values, it’s not clear which initial assessments have a prognostic value.

Patients with improved depression symptoms had significantly higher education than those who did not improve. Education has previously been found to have a protective effect on PSD (see for example Whyte et al. 2004; Journal of the American Geriatric Society). Given that there is empirical and statistical evidence for the importance of this variable, why wasn’t it included in any of the models?

Reviewer 2 Report

In their article the authors aspire to predict depression in patients who suffered a stroke with the help of machine learning algorithms.

The article is well-written and has a clear structure. From the medical perspective, the methods as well as the introduction and the presentation of the results are sound. 

There is one big problem for me as a reviewer though: Due to the focus on machine learning algorithms it is difficult - if not impossible - for me as a medical professional to check and verify their approach and methods thoroughly. The majority of the manuscript might be easier to review for someone specializing in computer sciences or mathematics. As far as I am able to evaluate this, there are no major concerns.

I would certainly recommend to change or remove the section beginning with "In our study, some cognitive impairments [...] were closely related to improvements in patients with PSD [...]" (p. 16, 426...). It is true that they call for "further studies" and additionally the limitations concerning multiple comparisons might not apply for machine learning. However, it is not possible to draw conclusions for patients from singular items in this study besides for the use in ML algorithms - if they would like to do that, adjustments regarding multiple comparisons would have to be used.

Reviewer 3 Report

The manuscript focusses on the impact of PSD on functional and cognitive improvement of the stroke patients. The area of the research is novel and interesting. The authors have put a lot of effort in the experimental protocol, which has led to the enhancement of the quality of the manuscript. 

After going through the manuscript, I have certain comments/queries as follows.

  1. What are the independent variables or input features in the input dataset? What is the dimension of the raw dataset?
  2. Why was feature extraction not considered necessary, before feature selection?
  3. Mann-Whitney U Test checks for any difference in the dependent variables of two independent groups. Please provide a brief explanation of how the method was used here for selecting or ranking features? Also, which technique was used- 'compare means' or 'compare medians'?
  4. To enhance the performance of the ML algorithm, features were dropped one at a time and checked. This process has been termed as wrapper method. However, feature selection techniques are categorized into wrapper-based, filter-based, etc. Hence, the idea is not clear. Please justify. 
  5. In section 2.3, line 148 is not clear. Do you mean 'one was used as test set?' Also, please review line 149- 'For each test set, the rest..........validation set.'
  6. As stated in line 299-301, how were the 18 and 6 parameters defined?
  7. Line 325- On what basis, the 5 parameters were considered the most important ones? Were the algorithms also executed using more than 5 parameters? If so, what were the accuracies obtained for different number of parameters?
  8. In section 3.4, the objective of the decision-making model is not clear.
  9. The authors have considered SVM_R as the best model for prediction based on the AUC with 5-fold cross-validation. However, there are other algorithms that exhibited better accuracies and sensitivities. Why only AUC was emphasized? Again, in lines 444-448, the authors have stated that SVM_R might not accurately detect PSD due to inadequate specificity. Please justify the statements. 
  10. What were the hyperparameters chosen for the algorithms? Did the authors implement any hyperparameter tuning? A hyperparameter tuning might end up into a more accurate prediction.

Round 2

Reviewer 1 Report

The authors have addressed my comments comprehensively. I believe the manuscript is now publishable in the current format.

Author Response

Thank you for your consideration!

Reviewer 3 Report

The authors have satisfactorily answered most of my queries and have incorporated substantial amendments in the revised manuscript. However, with reference to the last response, I feel that the authors might consider including details of different hyper-parameter sets (such as kernel types, C and gamma, etc.) used in the study along with the corresponding accuracy, preferably in a tabular form, in the manuscript or as supplementary information.

Author Response

Dear Reviewer 3,

Thank you for your consideration. We have answered your last question and the changes are marked in RED color in the 2nd revised manuscript.

Kind regards,

Jung Keun Hyun, M.D., Ph.D.

Point 1: The authors have satisfactorily answered most of my queries and have incorporated substantial amendments in the revised manuscript. However, with reference to the last response, I feel that the authors might consider including details of different hyper-parameter sets (such as kernel types, C and gamma, etc.) used in the study along with the corresponding accuracy, preferably in a tabular form, in the manuscript or as supplementary information.

Response 1: As this reviewer’s comment, we have added a table which included C and gamma values (scale and auto) in each kernel type (SVM_L and SVM_R) for the prediction of PSD occurrence and prognosis as a supplemental data (Supplemental table 3A and B), and explained in the result section as follows (Line 337-341).

“Accuracies of SVM_R with 10-fold cross-validation for the prediction of PSD occurrence and SVM_L with 5-fold cross-validation for the prediction of PSD prognosis were 7.000 and 0.771 respectively, and which were analyzed using hyper-parameters (Supplemental Table 3A and B).”

In addition, there was an error while moving the accuracy values of SVR_R with 5-fold and 10-fold cross-validations in Table 9 of the first revised manuscript, so it was corrected (Line 28 in the abstract, and Table 9, marked in RED color).